# Docosahexaenoic and Arachidonic Acids as Neuroprotective Nutrients throughout the Life Cycle

**DOI:** 10.3390/nu13030986

**Published:** 2021-03-18

**Authors:** Verónica Sambra, Francisca Echeverria, Alfonso Valenzuela, Raphaël Chouinard-Watkins, Rodrigo Valenzuela

**Affiliations:** 1Department of Nutrition, Faculty of Medicine, University of Chile, Santiago 8380000, Chile; vero.sambrav@uchile.cl (V.S.); franciscaecheverria@uchile.cl (F.E.); 2Faculty of Medicine, School of Nutrition, Universidad de Los Andes, Santiago 8380000, Chile; avalenzu50@gmail.com; 3Department of Nutritional Sciences, Faculty of Medicine, University of Toronto, Toronto, ON M5S1A8, Canada; raphael.chouinard.watkins@utoronto.ca

**Keywords:** docosahexaenoic acid, arachidonic acid, neuroprotection, neurodegeneration, Alzheimer’s disease, Parkinson’s disease

## Abstract

The role of docosahexaenoic acid (DHA) and arachidonic acid (AA) in neurogenesis and brain development throughout the life cycle is fundamental. DHA and AA are long-chain polyunsaturated fatty acids (LCPUFA) vital for many human physiological processes, such as signaling pathways, gene expression, structure and function of membranes, among others. DHA and AA are deposited into the lipids of cell membranes that form the gray matter representing approximately 25% of the total content of brain fatty acids. Both fatty acids have effects on neuronal growth and differentiation through the modulation of the physical properties of neuronal membranes, signal transduction associated with G proteins, and gene expression. DHA and AA have a relevant role in neuroprotection against neurodegenerative pathologies such as Alzheimer’s disease and Parkinson’s disease, which are associated with characteristic pathological expressions as mitochondrial dysfunction, neuroinflammation, and oxidative stress. The present review analyzes the neuroprotective role of DHA and AA in the extreme stages of life, emphasizing the importance of these LCPUFA during the first year of life and in the developing/prevention of neurodegenerative diseases associated with aging.

## 1. Introduction

Over many years there has been growing interest about n-3 and n-6 long-chain polyunsaturated fatty acids (n-3 and n-6 LCPUFA), mainly related to their role in neural development and the prevention of neurodegenerative diseases. Lipids correspond to 60% of the dry weight of the mammalian brain, mostly in the form of phospholipids [1,2]. N-3 and n-6 LCPUFA are required for proper brain growth and development, specifically docosahexaenoic acid (C22:6n-3, DHA) and arachidonic acid (C20:4n-6, AA) [3]. DHA and AA represent approximately 25% of the total content of brain fatty acids [2,4] and are constituents of the lipids of cell membranes that form the gray matter [1,5]. DHA represent up to 90% of the total n-3 LCPUFA in the brain [6]. This LCPUFA can be obtained from its dietary precursor alpha-linolenic acid (C18:3n-3, ALA), which, after a complex process of enzymatic desaturation and elongation, is converted to DHA, or can be obtained from preformed DHA of marine dietary sources [7]. DHA is mainly found in the phospholipids of synaptic terminal membranes, and the vast majority of DHA is incorporated into the structure of phosphatidylcholine, phosphatidylethanolamine, and phosphatidylserine [7,8,9]. The structural properties of DHA, such as its chain length and high unsaturation degree, give flexibility and fluidity to the neuronal plasma membrane, thus facilitating the signal transduction process into these cells [10,11]. These structural properties have a pivotal role in neuronal growth, migration, synaptogenesis, and synaptic plasticity [11,12,13]. In advanced stages of life, a decrease in plasma DHA levels is positively correlated with normal brain aging in healthy elderly individuals and also in patients diagnosed with neurogenerative diseases [14,15,16] such as Alzheimer’s disease (AD) [17,18]. It has also been observed that populations having a higher average dietary intake of DHA show a lower risk of developing cognitive impairment or AD [19,20]. Furthermore, a potential neuroprotective role of DHA in Parkinson’s disease (PD) through the increase in dopaminergic neurotransmission and the prevention of neuronal death is also actually recognized [21,22,23]. In this regard, the higher intake of n-3 LCPUFA (mostly DHA) trough supplementation may be effective to improve the nutritional status of the fatty acid and may play a role in maintaining brain health and in the prevention of brain aging symptoms [6,24,25].

AA is one of the most abundant n-6 LCPUFA in the brain and has a critical role in brain growth, being essential during the first months of life [26]. Ideally, during brain development, AA should be provided preformed in the diet because the low activity of the enzymes (Δ-5 and Δ-6 desaturases) involved in its formation from its dietary precursor linoleic acid (C18:2n-6, LA) results in a low conversion of LA to AA [27,28]. This fatty acid participates in many signaling pathways involved in cell division [29]. It is also a direct precursor of adrenic acid (C22:4n-6, AdA), a fatty acid necessary for neuronal development and enrichment of myelinic lipids [30,31]. A potential protective role of AA in neuronal aging through its participation in preserving the fluidity of the hippocampal cell membranes has been described [32,33,34,35]. However, opposite to this effect, a controversial role has been associated with AA in neuroinflammation and the genesis of cellular damage due to the pro-inflammatory action of some of AA metabolic derivatives [36,37,38]. According with this background, the present review analyzes the neurological role of DHA and AA in the extreme stages of life, emphasizing the importance of these LCPUFA during the first year of life and in the developing brain and in a context of age-related neurodegenerative disease prevention. Also, we discuss the eventual neuroprotective role of n-3 and n-6 LCPUFA in AD and PD.

## 2. DHA and AA: Biosynthesis, Metabolism, and Dietary Sources

ALA and LA are considered essential fatty acids because humans cannot synthesize them, so they must be supplied by the diet [39,40]. ALA is metabolized to eicosapentaenoic acid (C20:5 n-3, EPA) and subsequently to DHA, while LA is the precursor of AA [40]. ALA and LA are competitors in their respective metabolic pathways because both fatty acids are substrates for the same desaturase and elongase enzymes. In the synthesis of n-3 and n-6 LCPUFA in addition to the substrates ALA and LA, the participation of a complex enzymatic process which allows the desaturation and elongation of the 18 carbon atoms precursors (ALA and LA) is necessary [41]. This process occurs in microsomes and involves Δ-6 fatty acid desaturase 2 (FADS2), Δ-5 fatty acid desaturase 1 (FADS1), elongase 2 (ELOVL2) and elongase 5 (ELOVL5) enzymes [41]. The latter enzyme having a higher affinity for n-3 PUFA than for n-6 PUFA, favoring the transformation of ALA into DHA [42]. This less affinity of LA than ALA to desaturase and elongase enzymes has contributed to the recommendation of a 5:1 molar ratio for the dietary intake of n-6:n-3 PUFA, because when the dietary contribution of both fatty acids is similar, the formation of DHA is privileged compared to AA [43,44]. An interesting aspect about LCPUFA synthesis is the existence of polymorphisms in desaturase enzymes, which can influence both the ability to the synthesis of DHA or AA and the blood levels of these fatty acids [45]. For example, rs3834458 single nucleotide polymorphism in FADS2 may result in lower Δ-6 desaturase activity leading to higher ALA and lower DHA blood concentrations [46]. The presence of these polymorphisms can also influence lower levels of LCPUFA in breast milk, especially DHA, which could have impact on brain development [47,48,49]. The existence of polymorphisms has been associated with increased risk of developing insulin resistance, type 2 diabetes, cardiovascular disease, and non-alcoholic fatty liver disease (NAFLD), among other pathologies [50,51].

ALA is found in higher amounts in vegetable oils such as chia and flaxseed oil and in lower amounts in canola and soybean oils [52,53] The principal dietary sources of preformed DHA are fatty fish or blue fish, such as salmon, tuna, anchovy, sardine, and horse mackerel (Figure 1) [54,55,56]. The main dietary sources of LA are sunflower, soybean, and corn oils [52]. In contrast, the main dietary sources of AA are foods of animal origin, such as as beef, pork, lamb, chicken, turkey, and eggs [44]. Dietary sources of n-6 and n-3 polyunsaturated fatty acids influence brain development.

## 3. DHA and AA during Pregnancy and Breast Milk Period

During pregnancy and the first years of life, human beings have specific nutritional needs, which must be met to ensure proper development and growth [57]. Evolutionary changes about the biological mechanisms involved in transferring DHA and AA from mother to fetus during pregnancy [58,59] and lactation are well documented in the literature [60,61]. The chemical characteristics of human milk composition and its balance of nutrients and energy supply make it the optimal food source for infants [57]. In this regard, during the lactation stage the breast milk the lipid content increases from colostrum milk (2.36 ± 1.17 g/dL) to mature milk (3.39 ± 1.24 g/dL) [62], colostrum contains more LA than mature transitional milk, while colostrum contains less ALA than mature transitional milk [62]. Also, as lactation progresses, the proportions of DHA and AA decrease [62].

Human milk composition is the standard model to define adequate intakes of nutrients during the early stages of development in infants aged 0–2 years [63]. An example of these requirements occurs with AA, which is not considered an essential fatty acid for a healthy adult because habitual diet provides LA in amounts greater than 2.5% of the total caloric value per day requirement [44]. However, for infants aged 0-6 months, AA must be supplied in the diet within a range of 0.2% to 0.3% of the total caloric value per day requirement, establishing the concentration of preformed AA present in maternal milk as an adequate intake criterion which has been reported since the 1970s [44,64]. Both DHA and AA are always present in human milk, although in a variable quantity that is mainly determined by the mother’s diet [65,66] and the period of lactation [67]. Brenna et al., 2007 [66] carried out a systematic meta-analysis of the fatty acid composition of human breast milk obtained from 106 studies carried out in 33 countries, including a total of 2474 women, reporting a mean concentration of 0.47% for AA and 0.32% for DHA. Furthermore, the same group reported less variability in AA levels in breast milk compared to DHA levels with a coefficient of variation of 29% and 69%, respectively [55], indicating a greater variation in DHA synthesis than AA synthesis. This result may be due, in part, to the fact that approximately 90% of AA present in maternal milk fat is not derived from the diet, but from maternal body reserves [68]. However, DHA content is directly related to the mother’s dietary intake of ALA and/or DHA and, therefore, to the higher or lower access to foods that are a source of n-3 PUFA or n-3 LCPUFA [66]. Another study carried out by Fu et al., 2016 [69] provided information about 78 studies from 40 countries, and from 4163 samples of breast milk, reporting mean AA and DHA levels of 0.55% and 0.37%, respectively. Stability of AA levels in human milk takes an important biological significance when preformed AA is provided to infants who cannot obtain the fatty acid by de novo synthesis, or because its synthesis from the LA precursor is insufficient due to the low activity of desaturase enzymes [27,28,70]. Dietary DHA and AA, or tissue reserves of these LCPUFA must be transported to the placenta or to the breast incorporated into phospholipids and lysophospholipids [71]. Transport problems, not directly related to dietary deficiencies, may reduce the metabolic availability of LCPUFA, thus affecting normal brain development [71,72]. It has been estimated that during the first six months of the infant’s life, the mother supplies through breastfeeding around 3.9 kg of total lipids which is equivalent to 35,000 kcal [67]. This quantity is based on an average intake of infant lipids of 21.42 g/day, which provides 190 mg/day of AA and 130 mg/day of DHA [63,70]. These estimations were made using an average consumption of 850 mL/day of breast milk [73]. 

Traditionally the impact of LCPUFA intake during pregnancy and their effect in breast milk LCPUFA levels has been evaluated in normal-weight pregnant woman. However, the increase in obesity during pregnancy and breastfeeding as well as gestational diabetes, among other pathologies observed during the last decade, could alter the transport of LCPUFA to the placenta leading to a decrease in LCPUFA (mainly DHA) availability [72,74,75]. Currently, the potential impact in brain development of the transport of LCPUFA to the fetus or to the infant in maternal obesity and/or gestational diabetes, is a main study objective [71,75], findings that justify LCPUFA intake recommendations during pregnancy and lactation [71,75,76].

## 4. DHA and AA in Brain Development

The mammalian brain is a heterogeneous structure in which the maturation of the different brain regions varies over time [77]. The development of the human brain begins with the formation of the primitive neural tube during the first four weeks of gestation. The proliferation and cellular differentiation of neurons and glial cells (mainly astrocytes and oligodendrocytes) occurs between 5 and 18 weeks [77,78], a process in which are formed up to 200,000 neurons per minute [79,80]. Parallel to the multiplication of neurons, an active neuronal migration process occurs where glial cells act as auxiliary structures that guide neurons during their migration, facilitating neuronal mobility from the ventricular (central) germinal zone to the peripheral areas of the developing brain (neocortex) [81]. This is a transcendental event since migration defines the place where the neurons will be finally positioned in the different brain segments, thus determining the synaptic connection patterns that will be established and their future functionality [82,83]. Myelination begins during the third trimester of gestation, an active process that may extend until the third decade of life [2,77]. Also, during the last trimester of gestation, a significant increase in the number of synapses occurs. Neuronal cells become arborized and stratified in a laminar pattern that favors the connection, and up to 200 billion synapses can be formed [84]. This process is followed by programmed apoptosis; about 50% of neurons undergo programmed cell death, allowing the surviving neurons differentiation and specialization [82]. An explanation of this apoptotic phenomenon has been proposed; it would be a corrective mechanism to avoid erratic neuronal migration and/or due to the lack of trophic stimuli for some neurons that survive [11].

In humans, functional specialization of neurons occurs early from gestation and may even extend into early adulthood [83,85]. To quantify the stages of growth of the human brain, Dobbing and Sands (1973) analyzed 139 whole human brains of ages between 10 weeks of gestation until 7 years of life, and 9 adult brains [86]. The authors reported that the most significant proliferation of neuronal cells occurs during the period between 10 to 18 weeks of gestation, a period where may even reach the levels of neuronal proliferation that will be acquired in adulthood. Also, during the last trimester of pregnancy and the first two years of life, a rapid accumulation of both DHA and AA occurs in the developing brain [86]. In line with those results, Remko S.Kuipers et al. (2012) [87], determined that western infants accumulate fatty acids in the brain in the order AA> DHA > LA during all the stages of pregnancy, reaching the highest fatty acid accretion rate during the last five weeks of gestation [87]. During the second year of life, brain growth and neuronal structural reorganization continue rapidly [88]. The gray matter of the brain, which accumulates the greatest amount of DHA, increases in volume up to 5 years old [89]. It is known that 40–45% of the total lipids of the adult brain correspond to LCPUFA, with DHA and AA representing the vast majority (DHA 35–40% and AA 40–50%) (Figure 1) [11].

A study carried out by Lepping et al. (2019) [90] evaluated if supplementing with n-3 and n-6 LCPUFA during the first year of life could influence neurodevelopment. Newborns between 1 and 9 days old and up to 12 months old were randomly assigned to one of four terminal infant formula based on cow’s milk that had the same amounts of nutrients and ingredients except for LCPUFA and supplemented as follows: (i) control formula without DHA or AA; (ii) formula with 0.32% of fatty acids as DHA (17 mg/100 kcal); (iii) formula with 0.64% of fatty acids as DHA (34 mg/100 kcal); and (iv) formula with 0.96% of fatty acids as DHA (51 mg/100 kcal) [90]. All formulas containing DHA also provided 0.64% of fatty acids as AA (34 mg/100 kcal). The sources of DHA and AA were unicellular algae oils (*Crypthecodinium cohnii*) and fungi (*Mortierella alpina*), respectively [90]. The content of the other main fatty acids, including LA (16.9–17.5% of total fatty acids) and ALA (1.61–1.68% of total fatty acids), was similar for the four formulas; this study showed that the LCPUFA supplementation during infancy has lasting effects on brain structure, function, and neurochemical concentrations in regions associated with attention (parietal) and inhibition anterior cingulate cortex (ACC) [90]. Birch et al. (2010) [91] studied newborns that were randomly assigned to consume a formula that did not contain LCPUFA (control) or a formula where AA was 0.64% of total fatty acids and with variable amounts of DHA (0.32%, 0.64%, or 0.96% of total acids fatty) from birth until 12 months old [91]. At the age of 9 years, it was observed through functional magnetic resonance spectroscopy that there was greater brain activation in the ACC and parietal regions in children supplemented with n-3 LCPUFA [91]. The analysis showed that children belonging to the 0.64% DHA group exhibited greater connectivity between the prefrontal and parietal regions than all other groups [91]. Furthermore, a voxel-based analysis revealed that groups receiving 0.32% and 0.64% DHA had higher white matter volume in the ACC and parietal regions than controls and the 0.96% DHA group [90]. Thus, n-3 LCPUFA supplementation during childhood has long-lasting effects on brain structure, function, and neurochemical concentrations in regions associated with attention such as parietal) and ACC [90]. DHA/AA balance is an important variable to consider in the contribution of LCPUFA to cognitive and behavioral development in childhood. However, it has been described that blood DHA levels generally increase when increasing DHA supplementation, although these levels tend to stabilize when supplementation exceeds 0.64% [90]. This can contribute to explain why the infants fed the 0.96% DHA supplementation did not have more brain connectivity than the ones in the 0.64% DHA group [92]. AA levels showed a strong inverted U function in response to increased DHA supplementation when neural development was evaluated, i.e., at the highest DHA containing formula (0.96%) (DHA:AA ratio 1.5:1.0); it was observed that AA decreased in the blood, as compared to the two low and intermediate formulas (0.32% and 0.64% DHA) [92]. Today is a standard procedure the fortification of infant formulas with DHA and EPA, which are added as triglycerides, phospholipids, or fatty acid concentrates [76].

## 5. DHA and AA in Neuronal Development and Function

DHA and AA can modulate neuronal function by influencing: (i) the physical properties of neuronal membranes by modulating ion channels and vesicular transport for endo/exocytosis of membrane-bound proteins [11,93]; (ii) signal transduction, by modulating G protein-mediated second messenger systems; and (iii) gene expression, through direct binding to transcription factors [94] or through the regulation of signaling cascades by eicosanoids derived from AA and DHA-derived docosanoids. In this sense, DHA and AA are crucial for the metabolism, growth, and differentiation of neurons [11,95] (Figure 2).

### 5.1. DHA in Neuronal Metabolism

DHA represents more than 90% of the n-3 LCPUFA in the brain [31], mainly as part of the membrane phospholipids in the brain gray matter [6,96], constituting 35% of the total fatty acids in the synaptic membranes [97]. DHA is primarily esterified to phosphatidylethanolamine, phosphatidylserine, and to a lesser extent phosphatidylcholine in neuronal membranes. The structural properties of DHA, such as the length of its carbon chain and its six double bonds, give flexibility and fluidity to the neuronal plasma membrane, facilitating signal transduction into the cell [10,11]. The fluidity of the neuronal membrane facilitates the lateral movement of receptors, G proteins, ion channels [98], enzymes [99], and neuroreceptors, increasing the efficiency in signal transduction [100,101,102]. In the brain, DHA is involved in neuronal growth, neuronal migration, synaptogenesis, synaptic plasticity, and gene expression [11,12,13,103]. Neurons cannot form DHA from its precursor ALA, but the glial cells, (especially astrocytes) can desaturate and elongate dietary ALA to convert it into DHA, which is subsequently transferred to most neurons [104,105]. DHA can also modulate the expression of genes related to neuronal energy generation involved in the function of the respiratory chain and ATP synthesis (adenosine triphosphate synthase) [103,104,105]. That function is relevant because approximately 50% of mitochondrial ATP is consumed by the Na+/K+ ATPase pump to maintain cell homeostasis and ionic gradients, a fundamental requirement for neuronal electrical excitability [103,104,105].

### 5.2. AA in Neuronal Metabolism

As already mentioned, AA is one of the most abundant fatty acids in the brain [26]. This n-6 LCPUFA is indispensable for brain growth and modulation of cell division and signaling [29]. During brain development, the concentration of AA increases rapidly [106]. It has been described in animal models that approximately 70–80% of AA concentration that is reached in adulthood is the result of its cerebral accumulation in the early postnatal period [2]. AA is an immediate precursor of adrenic acid (C22:4n-6, AdA), fatty acid found in large amounts in myelinic lipids, especially in phosphatidylethanolamine and phosphatidylcholine [106,107], suggesting the fundamental role of AA as a precursor of AdA in the development of neural tissue [106]. Conversion of AA to AdA may represent an important mechanism for supplying the high demand for AdA at the brain level, which is essential for neuronal myelinic lipid enrichment [2,106]. Wijendran et al. (2002) [108] investigated the metabolism of preformed AA in newborn baboons by the administration of a single oral dose of C^13^-labeled AA, reporting that 79–93% of AA consumed accumulates in brain membrane lipids and approximately 5% to 16% of AA is transformed into AdA [108]. Brain accumulation of AA and AdA occurs during the first month of life and represent 17% and 8% of the total n-6 LCPUFA, respectively [108].

Another brain function of AA is directly related to its participation in phosphatidylcholine (PC) structure [107]. Some intracellular phospholipid bilayers include AA-containing PC (AA-PC), a structure that plays a role as second messenger participating in the long-term enhancement of synapses in the CA1 region of the hippocampus [109]. Using image mass spectrometry, Yang et al. (2012) [110] characterized the distribution of AA-PC within neurons in cultured upper cervical nodes, finding an increasing gradient of AA-PC along the proximal to distal axonal axis suggesting that this structure is an important source of free AA [110]. Furthermore, it has been described that free AA can activate protein kinases and ion channels and inhibit neurotransmitter recycling [29], thus contributing to better control of synaptic transmission [70]. In addition to the role of AA in modulating neuronal excitability, AA is also essential in neuronal development in part because it is directly responsible for the activation of syntaxin-3, a protein of the neuronal membrane involved in the growth and neurite repair, an essential process in neurogenesis and subsequently in synaptic transmission [111].

### 5.3. DHA and AA as Precursors of Endocannabinoids

Remarkably, LCPUFA levels in the brain are highly correlated with dietary intake of PUFA and LCPUFA [112]. DHA and AA are precursors of many bioactive lipid mediators identified as docosanoids and eicosanoids, respectively, which are actively involved in regulatory responses in inflammation (eicosanoids such as prostaglandins, prostacyclins, thromboxanes, leucotrienes) and in resolution of inflammation (docosanoids such as resolvins and maresins) [113]. Both LCPUFA are also involved in the formation of endocannabinoids (eCB) with regulatory effects at the central nervous system (CNS) [114]. The endocannabinoid system consists of eCB, eCB receptors CB1 and CB2, and associated anabolic and catabolic enzymes [115]. Their functions are to maintain body energy homeostasis through the nutrient availability detection and the modulation of orexinergic inputs in selective regions of the CNS [116]. PUFA derived eicosanoids and eCB have been identified as independent ligands of CB1 and CB2 receptors [115]. Hammels et al. (2019) [115] reported that CB1 ligands synthesis in CNS depends on dietary intake of AA and DHA in a model of fatty acid desaturase 2-deficient mouse [115]. Moreover, DHA and AA have been recognized as ligands of the nuclear RxR receptor in the brain, being the PUFA ratio in the western diet, a critical nutritional parameter for numerous neurodegenerative diseases [115], which could increase neuroinflammation and over-stimulation of the endocannabinoid system [117]. AA bound to phospholipids determines the formation of eCB, anandamide (AEA) and 2-arachidonoylglycerol (2-AG), molecules involved in the regulation of neuroinflammatory responses by microglia and astrocytes [114]. On the contrary, long-term supplementation with DHA and EPA reduces AEA and 2-AG synthesis [118]. N-3 and n-6 PUFA derived eCB are synthesized by lipoxygenases (LOX), cyclooxygenase 2 (COX-2), and cytochrome P450 epoxygenases (CYP450). Nevertheless, only LOX and CYP450 metabolites have been reported for a DHA eCB derivative; n-docosahexaenoylethanolamide (DHEA), and only CYP450 metabolites for an EPA eCB derivative; eicosapentaenoyl ethanolamide (EPEA) [116]. The physiological role of the metabolites generated by LOX and CYP450 remains to be elucidated to understand how these derivatives modulate cell signaling in health and disease [116]. A better understanding of the relationship between DHA, AA, and the endocannabinoid system is expected to lead to advances in the development of their therapeutic potential and the development of more specific treatment options for the prevention and treatment of neurodegenerative diseases [114].

## 6. DHA and AA in Neuroprotection

The most frequent neurodegenerative age-related diseases are Alzheimer’s Disease (AD) and Parkinson’s Disease (PD) [119]. Although the etiopathogenesis and clinical characteristics of AD and PD are different, these diseases share common mechanisms of damage, such as mitochondrial dysfunction [120], neuroinflammation [121], and oxidative stress [122]. AD is characterized by dementia, memory loss, and cognitive decline, disorders that are worsened with aging [123]. In 2019, the World Alzheimer Report indicated that more than 50 million people live with dementia worldwide, a number that will increase up to 152 million by 2050 [124]. In this regard, every three seconds, a person develops dementia, and the current annual cost of dementia is estimated at US $ 1 trillion, which is estimated to double by 2030 [124].

DHA via enzyme 15-lipoxygenase (15-LOX) can be converted in oxylipins, including resolvins and neuroprotectins, which are potent lipid mediators [125,126]. Furthermore, DHA may undergo lipid peroxidation producing oxylipin metabolites, such as 4-hytoxyhexenal (4-HHE) [126,127]. These metabolites may have a role in modulating oxidative cell homeostasis by regulating the activity of the transcription factors nuclear factor kappa-B (NF-κB) and nuclear factor erythroid 2-related factor 2 (Nrf2), thus participating in the inflammatory and antioxidant response, and neuroprotection [126]. Osterman et al. (2019) [128] assigned healthy adults with low fish consumption (*n* = 121) to receive capsules with different doses of n-3 LCPUFA reflecting three patterns of fatty fish consumption: 1, 2, or 4 servings/week with 3.27 g of EPA + DHA (1:1.2) per serving or placebo. The authors reported that plasma oxylipins after 3 and 12 months increased linearly with the highest intake of EPA and DHA [128].

Aging is a normal process of the life cycle, and its progression is usually accompanied by the decrease of a wide range of body functions, including cognitive function, marked by decreased synaptic density, decreased neuronal survival, and loss of volume of the gray and white matter [105,129,130]. Alteration of lipid metabolism also occurs [131] and is associated with the dysfunction of fluidity and activity of brain cell membranes microdomains (or rafts) [132]. Exacerbation of alteration of lipid metabolism generates abnormal brain activity that can potentially lead to the development of neurodegenerative diseases [132,133]. Factors contributing to the early cognitive decline include diseases related to unhealthy lifestyles and metabolic syndrome [104]. Among them, atherosclerosis and hypertension are relevant because an altered blood flow generates hypoperfusion and vascular dysfunction, causing an impairment of the blood-brain barrier and triggering neurodegenerative processes that lead to cognitive deterioration and, ultimately, to the development of AD [131,132].

The pathogenesis of AD has not been fully resolved. However, it is known that the mechanisms that trigger the pathology are related to: (i) low levels of acetylcholine [134]; (ii) aggregation of insoluble β-amyloid (Aβ) peptides, a product of abnormal processing of the amyloidal precursor protein, in neuritic plaques, leading to Aβ accumulation in the CNS [135,136,137]; (iii) hyperphosphorylation of the tau protein associated with microtubules, causing intraneuronal accumulation of neurofibrillary tangles of tau protein and disruption of neuronal microtubules [138,139]; and (iv) oxidative stress, leading to inflammation, synaptic toxicity and accumulation of intraneuronal inclusions [136,140] (Figure 3). Regarding its clinical characteristics, AD is a slowly progressive disease and three stages can be recognized in its evolution: the first is characterized by memory failures; in the second, language disorders, apraxias, and Gerstmann syndrome (agraphia and agnosia) are frequently added; and in the third stage, the patient is physically disabled and prostrated [141,142].

PD is the second most common neurodegenerative disease after AD [143], affecting 2%-3% of the population ≥65 years old [144]. The etiopathogenesis of the disease includes: (i) intracellular aggregation of α-synuclein causing the formation of Lewy bodies [145,146]; (ii) neuronal loss of the substantia nigra pars compacta (SNPc) leading to a marked striatal dopamine deficiency [147,148]; (iii) mitochondrial dysfunction, due to defects in the activity and incorrect assembly of complex I of the mitochondrial electron transport chain [149,150] (inhibition of mitochondrial complex I induces neuronal cell death in the SNPc leading to dopaminergic decrease) [147,149]; (iv) oxidative stress [147], the nigral dopaminergic neurons are particularly vulnerable to metabolic diseases and oxidative stress [151,152,153,154]; and (v) neuroinflammation [147,155,156] (Figure 4). The clinical evolution of PD is characterized by motor disturbances due to progressive nigrostriatal dopaminergic neurodegeneration that occurs in parallel with decreased levels of striatal dopamine, dopaminergic synapses, and the density of dendritic spines in mid-striated spinal neurons [157]. Other non-motor symptoms of PD include cognitive impairment [158] and gastrointestinal dysfunctions [159,160].

## 7. Dietary DHA in Alzheimer and Parkinson Diseases

Various mechanisms have been described by which DHA would exert a neuroprotective effect on AD. It has been described that populations having a higher intake of DHA show a lower risk of developing cognitive impairment or AD [20,21]. DHA can activate synapsin-1, a protein that promotes synapsis in excitatory glutamatergic neurons in the hippocampus, thus contributing to synaptic plasticity and improving cognitive function [161,162]. Furthermore, synapsin-1 is involved in the release of neurotransmitters, axonal elongation, and maintenance of synaptic contacts [163,164], whose synthesis and further phosphorylation are determined by the brain neurotrophic factor [158,160], which is increased by DHA, thus facilitating synaptic transmission [161]. On the other hand, DHA inhibits tau protein phosphorylation, preventing the intraneuronal microtubule disassembling and the neurofibrillary tangles accumulation [136]. DHA can also reduce neuronal Aβ load by preventing its formation [163,164,165] and accumulation [163,166], thus reducing the toxicity produced by Aβ [165] and neuronal apoptosis [167]. A recent study suggests that DHA, but not AA, can suppress the production of oligomeric Aβ (oAβ) and the oAβ-induced increase of phosphorylated cytosolic phospholipase A2, synthesis of inducible nitric oxide synthase, production of reactive oxygen species, and activation of tumor necrosis factor α in microglia cells [168]. DHA can modulate AA metabolism in oAβ-stimulated microglia by suppressing oxidative and inflammatory pathways and upregulating the antioxidative pathway involving Nrf2/heme oxygenase-1 [168]. Moreover, DHA can prevent Aβ-induced necroptosis of THP-1 cells via the receptor interacting protein kinase-1 and -3 (RIPK1/RIPK3) signaling pathway. DHA treatment (0.5 μM, 1 μM) restored migration of THP-1 monocytes which was impaired by Aβ25-35 [169]. Also, the anti-inflammatory effect of DHA derived docosanoids, i.e., resolvins, prevent brain damage adjacent to the inflammatory state [170,171,172]. The OmegAD study (2015) evaluated the effect of the supplementation with EPA (0.6 g/day) and DHA (1.7 g/day) on the production of resolvins for six months in subjects with AD [168]. Analysis of mononuclear cells from peripherical blood of patients showed that supplementation with EPA and DHA maintained the production of resolvins, preventing deleterious changes in cognitive function [170]. Understanding the mechanisms underlying the effects of DHA on microglia should contribute to developing a nutraceutical therapy with DHA for the prevention and/or treatment of AD [168].

Clinical trials have shown that DHA supplementation allows to preserve cognitive abilities in healthy individuals (learning, memory, and verbal fluency), while in AD results are still not so clear. It must be considered that (i) the age of the subjects, (ii) the state of evolution of the disease, (iii) the daily amount of DHA ingested as a supplement, and (iv) the time of intervention, are relevant to clearly identify the benefits of DHA supplementation during aging and AD [173,174].

DHA has also been linked to the maintenance of synaptic homeostasis and neuronal activity by modulating the expression of nuclear α-synuclein proteins in PD [11,123]. DHA produces an increase in the expression of brain trophic factors [161], including glial cell-derived neurotrophic factor (GDNF) [162,175], contributing to the prevention of glial cell dysfunction, which is related to the early appearance of neurodegenerative pathologies [176]. GDNF has a protective effect against dopaminergic injury in *substantia nigra* by promoting anti-inflammatory pathways linked to the inhibition of the expression of transcription factor Nuclear Factor-κB (NF-κB) [177]. On the other hand, the administration of DHA to humans and mice may increase (i) dopaminergic neurotransmission, (ii) synaptic membrane formation, and (iii) dendritic spine density, thus preventing neuronal death and motor symptoms, such as the disturbance and impairment of the motor and non-motor coordination which are characteristic in PD [21,22,175,178,179,180,181,182,183,184]. The beneficial effects of DHA interventions have been demonstrated even when the fatty acid is administered after the almost full development of PD symptoms in humans [172]. This DHA effect is an antagonist to the action of l-3,4-dihydroxyphenylalanine (l-DOPA), the drug most used in the treatment of PD, which compensates for the loss of dopaminergic cells by improving the synthesis of dopamine at the synaptic terminals [185] but does not prevent the degeneration of dopaminergic neurons and has no effect on non-motor symptoms, which are determinative in the quality of life of PD patients [186].

## 8. Dietary AA in Alzheimer and Parkinson Diseases

The role of AA in neurodegenerative pathologies, such as AD and PD, has not been fully elucidated. Some studies carried out on the effect of AA-enriched diets on AD mouse models observed that these diets could prevent cognitive dysfunction caused by abnormal processing of the amyloid precursor protein (APP), contributing to reducing the formation of insoluble Aβ in neuritic plaques [184,187]. In contrast, Amtul et al. (2012) reported an increase in Aβ production and deposition in mice exposed to an AA-enriched diet [188]. The contrast in the results may be due to the different AA doses used, since dietary AA content was ten times higher in the study conducted by Amtul et al., 2012 [184,187,188]. On the other hand, regarding the neuroinflammation of AD, it has been widely documented in the literature that AD patients overexpress the enzyme cyclooxygenase 2 (COX-2) in the cortex and the hippocampus [36,37,38,189], thus increasing the formation of pro-inflammatory prostaglandins PGD2 and PGE2 [190]. However, although AA is a substrate for COX-2 [178], currently, there is insufficient and nonconclusive evidence linking dietary AA with the appearance of neuroinflammation [191]. Regarding synaptic function, studies in mice indicate that dietary AA has positive effects on cognition and synaptic plasticity in aging animals [106,192], which may establish a positive relationship between abilities of spatial memory and AA content in the hippocampus [179,193,194]. On the other hand, Kotania S. et al. (2006) reported that the supplementation of 240 mg/day of AA and DHA produced a significant improvement in immediate memory and attention score in a group of patients with mild cognitive dysfunction and organic brain lesions but with no diagnosis of AD (no-AD) [195]. However, no significant differences were observed when compared to the AD group. A possible mechanism that would explain this improvement in cognitive function observed in control and AD patients proposes that AA would act at the post-synaptic level through the long-term enhancement of the synaptic transmission and the expansion of this at the pre-synaptic level by exerting a direct modulation of voltage-dependent potassium channels, thus stimulating exocytosis of neurotransmitters [35,196,197,198,199].

AA role in PD is controversial. Lakkapa et al. (2016) proposed that epoxyeicosatrienoic acids (EETs), which are AA metabolites, could constitute a therapeutic target for PD since they are widely distributed in the brain and have shown anti-inflammatory and antioxidant effects [200]. Concordantly, the same authors reported that EETs increased the expression of antioxidant enzymes and attenuated oxidative stress and inflammation in a *Drosophila* model of PD [201]. On the other hand, it is known that protein α-synuclein self-assembles into toxic β-sheet aggregates in PD, while it assembles into α-helical oligomers in healthy neurons [202]. Iljina et al. (2016) studied α-synuclein multimers conformation in the presence of AA in vitro. The authors reported that AA induces the α-helical assembling of α-synuclein, leading to lower neuronal damage than the β-sheet multimers formed without AA treatment [202].

On the contrary, AA is a ligand of the fatty acid-binding protein 3 (FABP3) [203], which has been described as an early biomarker of dementia and PD and highly expressed in dopaminergic neurons [203]. In the presence of AA, the fatty acid binds to AA inducing α-synuclein oligomerization and aggregation in Neuro-2A cells [203]. In line with that result, Julien et al. (2006) described that patients with PD had higher AA brain levels evaluated postmortem [204]. Finally, a meta-analysis was conducted in 2019 to assess the influences of dietary fat in PD and concluded that cholesterol and AA intake increase PD risk [205].

## 9. Conclusions

In recent years, substantial progress has been made in understanding the cellular and molecular mechanisms underlying the neuroprotective effect of DHA and AA in the extreme stages of life. The relevance of DHA and AA during the first months of life is widely accepted because these LCPUFA play a fundamental role in neural development. However, although DHA in advanced stages of life is associated with neuroprotective effects, both in healthy humans and in patients with AD or PD, the role of AA in these pathologies is still controversial. Analysis of currently available information suggests directing future research towards understanding the role of AA in aging and neurodegenerative pathologies. Another pending research challenge will be to establish the dose-response relationship for DHA in future treatments of patients with AD or PD since, according to these results, the use of DHA could be an alternative or a complement to the pharmacological treatment of these pathologies in the early stages, as well as a selective and promising agonist in the prevention of premature brain aging and its associated pathologies. The role of personalized medicine, in the near future, together with a better knowledge of bioavailability, pharmacodynamics, pharmacokinetic and metabolic effects of DHA and AA challenges toward the development of more effective forms to supply these LCPUFA in benefit of neuroprotection throughout the life cycle.

## Figures and Tables

**Figure 1 nutrients-13-00986-f001:**
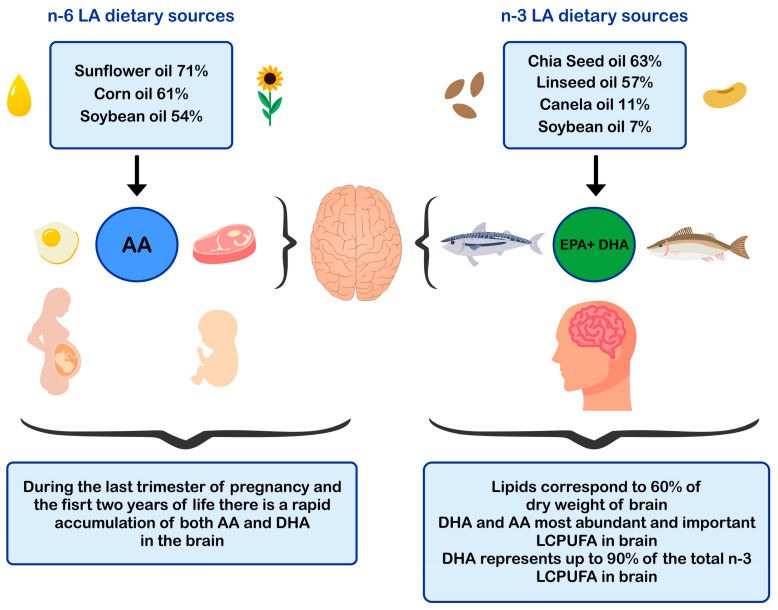
Dietary sources of n-6 and n-3 polyunsaturated fatty acids and impact in brain development. AA: arachidonic acid; EPA: eicosapentaenoic acid; DHA: docosahexaenoic acid; LCPUFA: long-chain polyunsaturated fatty acid; LA: linoleic acid.

**Figure 2 nutrients-13-00986-f002:**
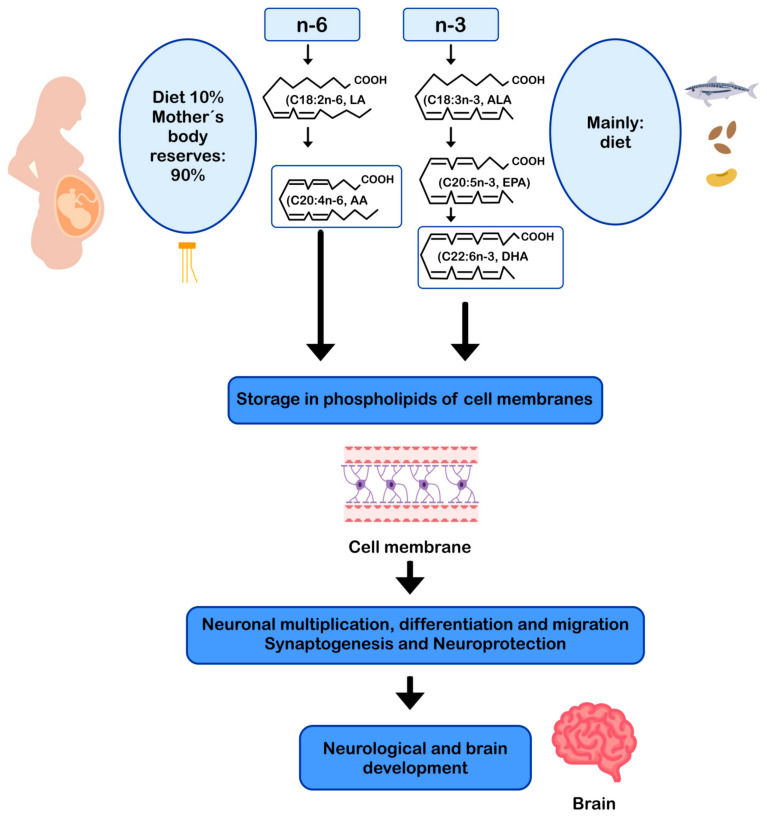
AA and DHA in neuronal development and function. AA: arachidonic acid; ALA: alpha-linolenic acid; DHA: docosahexaenoic acid; EPA: eicosapentaenoic acid; LA: linoleic acid.

**Figure 3 nutrients-13-00986-f003:**
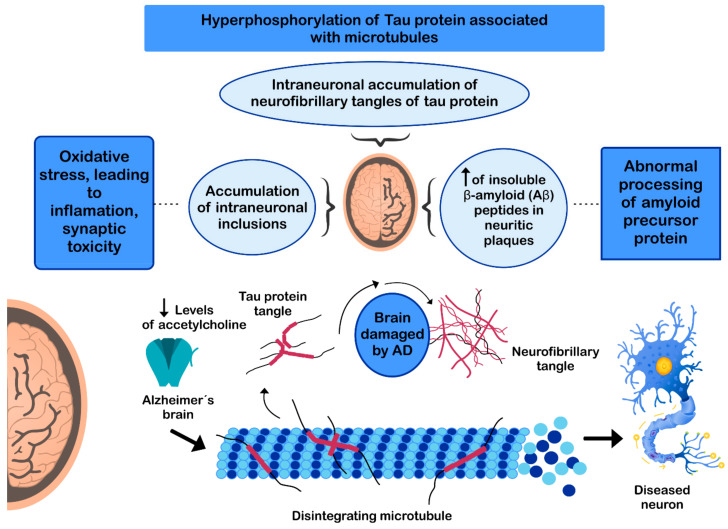
Mechanisms involved in the pathogenesis of Alzheimer’s disease (AD).

**Figure 4 nutrients-13-00986-f004:**
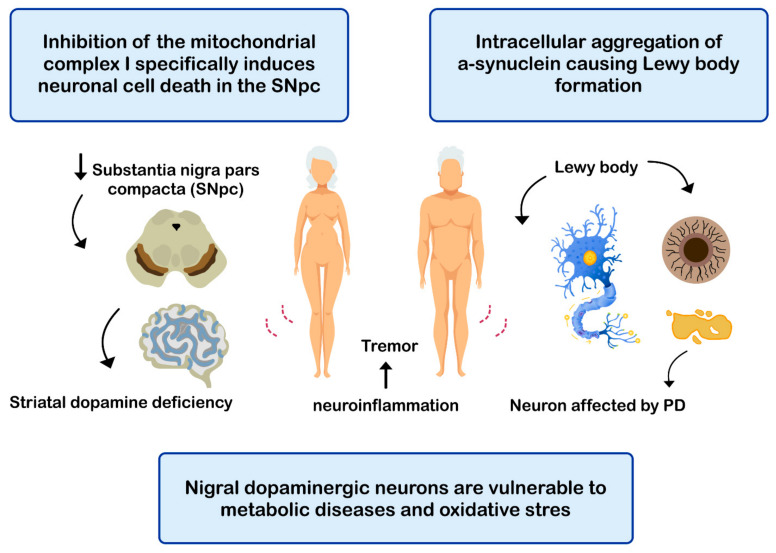
Mechanisms related with the pathogenesis of Parkinson’s disease (PD).

## Data Availability

Not applicable.

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
