# Peer review of "Docosahexaenoic and Arachidonic Acids as Neuroprotective Nutrients throughout the Life Cycle"

_nutrients, 2021, doi:10.3390/nu13030986_

Round 1

Reviewer 1 Report

The authors reviewed the study about the Docosahexaenoic and Arachidonic Acids as neuroprotective nutrients throughout the life cycle. The investigation of the n-3 PUFAS in neuroprotection has been widely studied. This article represents an important summary of the n-3 PUFAs related effects in neuroprotection.  

  1. Comments

  1. The authors presented a small introduction with very sparse neuroprotective related studies. We suggest the authors increase the introduction part and present the novelty of their study by clarifying their aim. Please check Cholewski M et al., 2018, Weiser MJ et al., 2016; Ajith TA et al., 2018; Derbyshire E et al., 2018; Lauritzen Lotte et al., 2016; Bianchi VE et al., 2019.
  2. The part of DHA and AA biosynthesis is lacking. There is a very wide metabolism description, authors did not mention related enzymes, deficiencies, despite the high ALA intake for instance there are different approaches that studied insufficient conversion to EPA and DHA. In addition, the 5 to 1 molar ratio definition is very descriptive.
  3. The authors used to mention the pregnancy content, but they did not focus on the transport differences and deficiencies in different related pathologies. Elshani B et al., 2021, Singh H, Thakur, et al., 2020.
  4. The authors summarized the DHA and AA in breast milk. Authors need to mention a very interesting systematic review regarding the content of breast milk which summarized the different stages of lactation as well Yang Ting et al., 2018. In addition, the authors did not focus on fortifying milk strategies with the EPA and DHA content.
  5. What about the brain development in pathological conditions with the inappropriate transfer of n-3 PUFAs from mother to child.Devarshi PP et al., 2019; See Sun GY et al., 2018; Mun JG et al., 2019, that focus on brain health and disease.
  6. Similarly, the authors focus on DHA and AA in neuronal development and function. This part is lacking and the figure is very general content and does not show too much explanation.  We suggest the authors to focus in Lauritzen L et al., 2016; Janssen Cl et al., 2014, Feltham BA et al., 2020; Carbone BE et al., 2020; Chizhikov D etc.
  7. Authors summarized studies in neuroprotection, however, different studies focused on this topic that summarized the role of DHA and AA in neuroprotection.  Cardoso C et al., 2016; Bazan NG et al., 2018; Zhu W et al., 2018; Freitas HR et al., 2018; Clementi ME et al., Lo Van A et al., 2016; Chiu HF et al., 2020;  Yip PK et al., 2019;  Hachem M et al., 2020; Oguro A et al., 2021.
  8. There are different systematic reviews about Alzheimer's and Parkinson's disease and the role of n-3 PUFAs. Authors need to enhance these parts with more evidence-based approaches. The role of DHA and AA need to correlate with the mechanism in Alzheimer's disease (the mechanism is not related to the main topic). A similar approach needs to be linked with Parkinson's disease rather than mentioning the general mechanism of the disease and mechanism which is still very superficial.
  9. I suggest the authors enhance their table more with the classification of clinical and preclinical findings, a period of study, clarification of concentrations, pharmaceutical forms of administration and increase their studies with additional evidence. Ajith TA. Et al., 2018;  Arellanes IC, et al., 2020;  Grimm MOW 2017 et al.; Yassine HN et al., 2017. Rolland Y, Barreto et al., 2019 etc. So please enhance the table and related studies to Alzheimer's disease, similarly with the Parkinson disease regarding both DHA and AA by dividing the experimental and clinical findings.
  10. Figure 5 is very general and such evidence is still controversial. The authors need to enhance Figure 5 with additional findings and explanations as per the previous comment in the tables.
  11. Again authors used to summarize some of the studies in the related pathologies in a similar approach Chiu HF, et al., 2020. However to better delineate their study, tables, and figures authors need to include every single study in the topic by not including the partial citation. We suggest the authors make the pro and contra-related studies in such pathologies. Moreover, authors need to add the future perspectives, the limitation of their study, and which are the ongoing strategies in order to increase such content, the possible role of personalized medicine and related pharmacogenetics that might affect the bioavailability, pharmacodynamics processes in neuroprotection, challenges in the pharmaceutical forms, etc.
  12. Authors used the studies with cognition process but very lacking in Alzheimer and Parkinson (table). So again authors need to include the additional studies playing role in neuroprotection and such neurodegenerative disease, phases, early or postpone protection, as prevention or treatment approach, etc.

Author Response

Thank you very much for reviewing our manuscript. Considering your comments we have made different changes to the text. We hope that these changes are sufficient. Again, we appreciate your help in improving the quality of our manuscript.

Reviewer #1

Observation

  1. The authors presented a small introduction with very sparse neuroprotective related studies. We suggest the authors increase the introduction part and present the novelty of their study by clarifying their aim. Please check Cholewski M et al., 2018, Weiser MJ et al., 2016; Ajith TA et al., 2018; Derbyshire E et al., 2018; Lauritzen Lotte et al., 2016; Bianchi VE et al., 2019.

Answer:

We added a new paragraph about the neuroprotective effects of LCPUFA in neurodegenerative diseases, especially for DHA.

New paragraphs:

Page 2.

“....In this regard, the higher intake of n-3 LCPUFA (mostly DHA) trough supplementation may be effective to improve the nutritional status of the fatty acid and may play a role in maintaining brain health and in the prevention of brain aging symptoms [6,24,25].”

Page 2.

 .....“Also, we discuss the eventual neuroprotective role of n-3 and n-6 LCPUFA in AD and PD.”

New references cited:

  1. Weiser MJ, Butt CM, Mohajeri MH. Docosahexaenoic Acid and Cognition throughout the Lifespan. Nutrients. 2016,17,8(2)–99
  2. Emma Derbyshire. Brain Health across the Lifespan: A Systematic Review on the Role of Omega-3 Fatty Acid Supple-ments. Nutrients. 2018,10(8),1094.
  3. Lauritzen L, Brambilla P, Mazzocchi A, Harsløf LB, Ciappolino V, Agostoni C. DHA Effects in Brain Development and Function. Nutrients. 2016,8(1),6.

  1. The part of DHA and AA biosynthesis is lacking. There is a very wide metabolism description, authors did not mention related enzymes, deficiencies, despite the high ALA intake for instance there are different approaches that studied insufficient conversion to EPA and DHA. In addition, the 5 to 1 molar ratio definition is very descriptive.

Answer:

As suggested, we added two new paragraphs about this comment.

New paragraphs:

Page 2.

...“ In the synthesis of n-3 and n-6 LCPUFA in addition to the substrates ALA and LA, the par-ticipation of a complex enzymatic process which allows the desaturation and elongation of the 18 carbon atoms precursors (ALA and LA) is necessary [41]. This process occurs in microsomes and involves Δ-6 fatty acid desaturase 2 (FADS2), Δ-5 fatty acid desaturase 1 (FADS1), elongase 2 (ELOVL2) and elongase 5 (ELOVL5) enzymes [41].”....

Page 2 to 3.

.....”An interesting aspect about LCPUFA synthesis is the existence of polymorphisms in desaturase enzymes which can influence both the ability to the synthesis of DHA or AA and the blood levels of these fatty acids [45]. For example, rs3834458 single nucleotide polymorphism in FADS2 may result in lower Δ-6 desaturase activity leading to higher ALA and lower DHA blood concentrations [46]. The presence of these polymorphisms can also influence lower levels of LCPUFA in breast milk, especially DHA, which could have impact on brain development [47,48,49]. The existence of polymorphisms has been associated with increased risk of developing insulin resistance, type 2 diabetes, cardiovascular disease and non-alcoholic fatty liver disease (NAFLD) among other pathologies [50,51]”.....

New references cited:

  1. Gonzalez-Soto M, Mutch DM. Diet Regulation of Long-Chain PUFA Synthesis: Role of Macronutrients, Mi-cronutrients, and Polyphenols on Δ-5/Δ-6 Desaturases and Elongases 2/5. Adv Nutr, 2020, 13,142.
  2. Brayner B, Kaur G, Keske MA, Livingstone KM. FADS Polymorphism, Omega-3 Fatty Acids and Diabetes Risk: A Sys-tematic Review. Nutrients. 2018,13,10(6),758.
  3. Chen X, Wu Y, Zhang Z, Zheng X, Wang Y, Yu M, Liu G. Effects of the rs3834458 Single Nucleotide Polymorphism in FADS2 on Levels of n-3 Long-chain Polyunsaturated Fatty Acids: A Meta-analysis. Prostaglandins Leukot Essent Fatty Acids. 2019,150,1-6.
  4. Xie L, Innis SM. Genetic variants of the FADS1 FADS2 gene cluster are associated with altered (n-6) and (n-3) essential fatty acids in plasma and erythrocyte phospholipids in women during pregnancy and in breast milk during lactation. J Nutr. 2008,138(11),2222-8.
  5. Steer CD, Davey Smith G, Emmett PM, Hibbeln JR, Golding J. FADS2 polymorphisms modify the effect of breastfeeding on child IQ. PLoS One. 2010,13,(7),e11570.
  6. Miliku K, Duan QL, Moraes TJ, Becker AB, Mandhane PJ, Turvey SE, Lefebvre DL, Sears MR, Subbarao P, Field CJ, Azad MB. Human milk fatty acid composition is associated with dietary, genetic, sociodemographic, and environmental factors in the CHILD Cohort Study. Am J Clin Nutr. 2019,1,110(6),1370-1383.
  7. Koletzko B, Reischl E, Tanjung C, Gonzalez-Casanova I, Ramakrishnan U, Meldrum S, Simmer K, Heinrich J, Demmelmair H. FADS1 and FADS2 Polymorphisms Modulate Fatty Acid Metabolism and Dietary Impact on Health. Annu Rev Nutr. 2019,39,21-44.
  8. Xu Y, Zhao Z, Liu S, Xiao Y, Miao M, Dong Q, Xin Y. Association of Nonalcoholic Fatty Liver Disease and Coronary Artery Disease with FADS2 rs3834458 Gene Polymorphism in the Chinese Han Population. Gastroenterol Res Pract. 2019,2019,6069870.

  1. The authors used to mention the pregnancy content, but they did not focus on the transport differences and deficiencies in different related pathologies. Elshani B et al., 2021, Singh H, Thakur, et al., 2020.

Answer:

Thanks for this comment; we added a brief new paragraph:

New paragraph

Page 4.

.....“Dietary DHA and AA, or tissue reserves of these LCPUFA must be transported to the placenta or to the breast incorporated into phospholipids and lysophospholipids [71]. Transport problems, not directly related to dietary deficiencies, may reduce the metabolic availability of LCPUFA thus affecting normal brain development [71,72].”

New references cited:

  1. Brikene Elshani, Vjosa Kotori, Armond Daci. Role of omega-3 polyunsaturated fatty acids in gestational diabetes, ma-ternal and fetal insights: current use and future directions. J Matern Fetal Neonatal Med. 2021,34(1),124-136.
  2. Álvarez D, Muñoz Y, Ortiz M, Maliqueo M, Chouinard-Watkins R, Valenzuela R. Impact of Maternal Obesity on the Metabolism and Bioavailability of Polyunsaturated Fatty Acids during Pregnancy and Breastfeeding. Nutrients. 2020,13(1),19.

  1. The authors summarized the DHA and AA in breast milk. Authors need to mention a very interesting systematic review regarding the content of breast milk which summarized the different stages of lactation as well Yang Ting et al., 2018. In addition, the authors did not focus on fortifying milk strategies with the EPA and DHA content.

Answer:

We appreciate these comments; We added the suggested references and two paragraphs regarding the contribution of LCPUFA in infant formulas.

New paragraphs

Page 3 to 4.

....”In this regard, during the lactation stage the breast milk the lipid content increases from olostrums milk (2.36 ± 1.17 g / dL) to mature milk (3.39 ± 1.24 g / dL) [62], olostrums contains more LA than mature transitional milk, while olostrums contains less ALA than mature transitional milk [62]. Also, as lactation progresses, the proportions of DHA and AA decrease [62].”

Page 6.

....“Today is a standard procedure the fortification of infant formulas with DHA and EPA, which are added as triglycerides, phospholipids or fatty acid concentrates [76].”....

New references cited:

  1. Yang T, Zhang L, Bao W, Rong S. Nutritional composition of breast milk in Chinese women: a systematic review. Asia Pac J Clin Nutr. 2018,27(3),491-502.
  2. Mun JG, Legette LL, Ikonte CJ, Mitmesser SH. Choline and DHA in Maternal and Infant Nutrition: Synergistic Implications in Brain and Eye Health. Nutrients. 2019,11(5),1125.

  1. What about the brain development in pathological conditions with the inappropriate transfer of n-3 PUFAs from mother to child. Devarshi PP et al., 2019; See Sun GY et al., 2018; Mun JG et al., 2019, that focus on brain health and disease.

Answer:

This interesting comment was addressed by the introduction of a new paragraph.

New paragraph

Page 4.

...“Traditionally the impact of LCPUFA intake during pregnancy and their effect in breast milk LCPUFA levels has been evaluated in normal-weight pregnant woman. However, the increase in obesity during pregnancy and breastfeeding as well as gestational diabetes, among other pathologies observed during the last decade, could alter the transport of LCPUFA to the placenta leading to a decrease in LCPUFA (mainly DHA) availability [72,74,75]. Currently, the potential impact in brain development of the transport of LCPUFA to the fetus or to the infant in maternal obesity and/or gestational diabetes, are a main study objective [71,75], findings that justify LCPUFA intake recommendations during pregnancy and lactation [71,75,76].”..

New references cited:

  1. Brikene Elshani, Vjosa Kotori, Armond Daci. Role of omega-3 polyunsaturated fatty acids in gestational diabetes, ma-ternal and fetal insights: current use and future directions. J Matern Fetal Neonatal Med. 2021,34(1),124-136.
  2. Álvarez D, Muñoz Y, Ortiz M, Maliqueo M, Chouinard-Watkins R, Valenzuela R. Impact of Maternal Obesity on the Metabolism and Bioavailability of Polyunsaturated Fatty Acids during Pregnancy and Breastfeeding. Nutrients. 2020,13(1),19.
  3. Grote V, Verduci E, Scaglioni S, Vecchi F, Contarini G, Giovannini M, Koletzko B, Agostoni C; European Childhood Obesity Project. Breast milk composition and infant nutrient intakes during the first 12 months of life. Eur J Clin Nutr. 2016.70(2),250-6.
  4. Barrera C, Valenzuela R, Chamorro R, Bascuñán K, Sandoval J, Sabag N, Valenzuela F, Valencia MP, Puigrredon C, Valenzuela A. The Impact of Maternal Diet during Pregnancy and Lactation on the Fatty Acid Composition of Erythro-cytes and Breast Milk of Chilean Women. Nutrients. 2018,10(7),839.
  5. Devarshi PP, Grant RW, Ikonte CJ, Hazels Mitmesser S. Maternal Omega-3 Nutrition, Placental Transfer and Fetal Brain Development in Gestational Diabetes and Preeclampsia. Nutrients. 2019,11(5),1107.
  6. Mun JG, Legette LL, Ikonte CJ, Mitmesser SH. Choline and DHA in Maternal and Infant Nutrition: Synergistic Implications in Brain and Eye Health. Nutrients. 2019,11(5),1125.

  1. Similarly, the authors focus on DHA and AA in neuronal development and function. This part is lacking and the figure is very general content and does not show too much explanation. We suggest the authors to focus in Lauritzen L et al., 2016; Janssen Cl et al., 2014, Feltham BA et al., 2020; Carbone BE et al., 2020; Chizhikov D etc.

Answer:

The section “DHA AND AA IN NEURONAL DEVELOPMENT AND FUNCTION” address the specific role of DHA and AA in brain development. Also, we improved figure 2.

  1. Authors summarized studies in neuroprotection, however, different studies focused on this topic that summarized the role of DHA and AA in neuroprotection. Cardoso C et al., 2016; Bazan NG et al., 2018; Zhu W et al., 2018; Freitas HR et al., 2018; Clementi ME et al., Lo Van A et al., 2016; Chiu HF et al., 2020; Yip PK et al., 2019;  Hachem M et al., 2020; Oguro A et al., 2021.

Comment:

Aspects related to the neuroprotective effects of DHA and AA are discussed in sections 6, 7 8 of the manuscript.

  1. There are different systematic reviews about Alzheimer's and Parkinson's disease and the role of n-3 PUFAs. Authors need to enhance these parts with more evidence-based approaches. The role of DHA and AA need to correlate with the mechanism in Alzheimer's disease (the mechanism is not related to the main topic). A similar approach needs to be linked with Parkinson's disease rather than mentioning the general mechanism of the disease and mechanism which is still very superficial.

Comment:

We included in the sections referred to “Neuroprotective……..” the most relevant aspects related to the protective mechanisms where DHA and AA are involved.

  1. I suggest the authors enhance their table more with the classification of clinical and preclinical findings, a period of study, clarification of concentrations, pharmaceutical forms of administration and increase their studies with additional evidence. Ajith TA. Et al., 2018; Arellanes IC, et al., 2020; Grimm MOW 2017 et al.; Yassine HN et al., 2017. Rolland Y, Barreto et al., 2019 etc. So please enhance the table and related studies to Alzheimer's disease, similarly with the Parkinson disease regarding both DHA and AA by dividing the experimental and clinical findings.

Answer:

To improve the understanding and the extention of the manuscript, we decided to delete Table 1. Also, we added a new paragraph.

New paragraph

Page 12.

....“ Clinical trials have shown that DHA supplementation allows to preserve cognitive abilities in healthy individuals (learning, memory and verbal fluency), while in AD results are still not so clear. It must be considered that i) the age of the subjects, ii) the state of evolution of the disease, iii) the daily amount of DHA ingested as a supplement, and iv) the time of intervention, are relevant to clearly identify the benefits of DHA supplementation during aging and AD [173,174]..”

New references cited:

  1. Yassine HN, Braskie MN, Mack WJ, Castor KJ, Fonteh AN, Schneider LS, Harrington MG, Chui HC. Association of Do-cosahexaenoic Acid Supplementation With Alzheimer Disease Stage in Apolipoprotein E ε4 Carriers: A Review. JAMA Neurol. 2017,74(3),339-347.
  2. Quinn JF, Raman R, Thomas RG, Yurko-Mauro K, Nelson EB, Van Dyck C, Galvin JE, Emond J, Jack CR Jr, Weiner M, Shinto L, Aisen PS. Docosahexaenoic acid supplementation and cognitive decline in Alzheimer disease: a randomized trial. JAMA. 2010,304(17),1903-11.

  1. Figure 5 is very general and such evidence is still controversial. The authors need to enhance Figure 5 with additional findings and explanations as per the previous comment in the tables.

Answer:

Considering this comment and the main aspects discussed in the manuscript. We decided to delete figure 5.

  1. Again authors used to summarize some of the studies in the related pathologies in a similar approach Chiu HF, et al., 2020. However to better delineate their study, tables, and figures authors need to include every single study in the topic by not including the partial citation. We suggest the authors make the pro and contra-related studies in such pathologies. Moreover, authors need to add the future perspectives, the limitation of their study, and which are the ongoing strategies in order to increase such content, the possible role of personalized medicine and related pharmacogenetics that might affect the bioavailability, pharmacodynamics processes in neuroprotection, challenges in the pharmaceutical forms, etc.

Answer:

Considering this comment, we added a new paragraph in the Conclusions section.

New paragraph

Page 13.

...“The role of personalized medicine, in the near future, together with a better knowledge of bioavailability, pharmacodynamics, pharmacokinetic and metabolic effects of DHA and AA challenges to the development of more effective forms to supply these LCPUFA in benefit of neuroprotection throughout the life cycle.”. 

  1. Authors used the studies with cognition process but very lacking in Alzheimer and Parkinson (table). So again authors need to include the additional studies playing role in neuroprotection and such neurodegenerative disease, phases, early or postpone protection, as prevention or treatment approach, etc.

Comment:

The main aspects related to the neuroprotective effects of DHA and AA are discussed in sections 6, 7 8 of the manuscript. Considering the extension of the text, we request not to include new texts and references.

Reviewer 2 Report

This is an excellent review of the role that DHA and AA have on neurogenesis and brain development throughout life. The emphasis is on the biochemistry and metabolism of the n-3 and n-6 LCPUFAs. There are a few typos that need to be addressed. Also, some important articles are missing  as indicated below:

  1. Page 2, para 3: "AL" is not defined. I think you mean LA.
  2. Page 3, Fig. 1: left lower box should read, During the last trimester...
  3. Page 6, Fig. 2: The images on the bottom of the figure are not described.

Author Response

Reviewer 2#

We really appreciate the comment of reviewer.

Comments:

This is an excellent review of the role that DHA and AA have on neurogenesis and brain development throughout life. The emphasis is on the biochemistry and metabolism of the n-3 and n-6 LCPUFAs. There are a few types that need to be addressed. Also, some important articles are missing as indicated below:

  1. Page 2, para 3: "AL" is not defined. I think you mean LA.

Answer:

We apologise by this error. We replaced “AL” by “LA”

  1. Page 3, Fig. 1: left lower box should read, During the last trimester...

Answer:

Thanks for this comment. We improved the text of figure 1.

  1. Page 6, Fig. 2: The images on the bottom of the figure are not described.

Answer:

According to this comment we improved figure 2.

Reviewer 3 Report

The review is well written and informative.  The authors accomplished great job.

I hope the authors will also mention AA cascade.

Author Response

Reviewer 3#

Comments:

The review is well written and informative. The authors accomplished great job.

I hope the authors will also mention AA cascade.

Answer:

We really appreciate the comment of reviewer.

We added a new paragraph about the AA cascade.

New paragraph

Page 8.

“DHA and AA are precursors of many bioactive lipid mediators identified as docosanoids and eicosanoids, respectively, which are actively involved in regulatory responses in inflammation (eicosanoids such as prostaglandins, prostacyclins, thromboxanes, leucotrienes) and in resolution of inflammation (docosanoids such as resolvins and maresins) [113].”

New reference:

  1. Buckley CD, Gilroy DW, Serhan CN. Proresolving lipid mediators and mechanisms in the resolution of acute inflam-mation. Immunity. 2014,40(3),315-27.